# Prospects of Cybersecurity in Smart Cities

Fernando Almeida [1,2]

1 Polytechnic Higher Institute of Gaya (ISPGAYA), 4400-103 Vila Nova de Gaia, Portugal; almd@fe.up.pt
2 Institute for Systems and Computer Engineering, Technology and Science, 4200-465 Porto, Portugal

**Abstract:** The complex and interconnected infrastructure of smart cities offers several opportunities for attackers to exploit vulnerabilities and carry out cyberattacks that can have serious consequences for the functioning of cities' critical infrastructures. This study aims to address this phenomenon and characterize the dimensions of security risks in smart cities and present mitigation proposals to address these risks. The study adopts a qualitative methodology through the identification of 62 European research projects in the field of cybersecurity in smart cities, which are underway during the period from 2022 to 2027. Compared to previous studies, this work provides a comprehensive view of security risks from the perspective of multiple universities, research centers, and companies participating in European projects. The findings of this study offer relevant scientific contributions by identifying 7 dimensions and 31 sub-dimensions of cybersecurity risks in smart cities and proposing 24 mitigation strategies to face these security challenges. Furthermore, this study explores emerging cybersecurity issues to which smart cities are exposed by the increasing proliferation of new technologies and standards.

**Keywords:** cybersecurity; smart cities; intelligent devices; cyber-attacks; privacy

## 1. Introduction

Smart cities are urban spaces characterized by the widespread use of Information and Communication Technologies (ICTs). They intend to improve political-economic efficiency and support human and social development, thus improving the quality of life of its citizens. According to Toli and Murtagh [1], a city is considered "smart" when there are drivers of sustainable eco-economic growth, a high quality of life, and conscious management of natural resources through participatory and democratic governance. Several drivers of smart cities include investments in technological infrastructure, investments in human capital, and social investments. In addition, other aspects that should be considered when classifying cities as smart include urban mobility, commitment to environmental issues, and social issues [2–5]. In summary, smart cities use technology and innovation to improve the quality of life of their inhabitants, increase the economic value generated in that community, and promote environmental sustainability.

Smart cities focus on the provision of a set of initiatives, actions, and services, in various areas of applicability in cities that aim to optimize and improve the well-being of their populations, both in terms of health and the environment. In broad terms, the goal of smart cities is to dynamically optimize the city to provide a set of actions and services for a more inclusive and sustainable city [6,7]. The main role that ICTs can assume is to integrate information systems services from each domain of activity of a city, such as health, education, transportation, energy, water, and waste management, to provide public services to citizens in a more efficient and ubiquitous way [8]. Furthermore, ICT plays a role in the incorporation and integration of various complex systems at the level of technological infrastructure, social structures, politics, and human behavior [9,10].

At the same time, the role of ICT for businesses, citizens, and civil society to create, invent, and experiment new things to improve and optimize the quality of life in cities is also widely known and accepted. However, taking into account the exposure of digital

services and support services of the various domains through communication networks and the use of ICT, it is important to consider the identification of risks and challenges associated with smart cities, to reduce or mitigate these risks, namely in different sectors of activity, such as industrial control systems, intelligent transportation systems, Internet of Things (IoT), and digital health (e-health), in addition to other areas of intervention of cities. The literature [11,12] recognizes that cybersecurity and data privacy issues have become increasingly relevant, questionable, and challenging, both for citizens and for technology companies that provide digital services who are aware of the problem of cybercrime and cyberterrorism.

Cybersecurity plays a key role in smart cities due to the increasing interconnectivity and widespread use of ICT [13]. In smart cities, systems and devices are networked to collect data, monitor infrastructures, optimize services, and improve citizens' quality of life. However, this connectivity exposes cities to a range of cyber threats, making data and infrastructure security critical. Systematic literature reviews performed by Kim et al. [14] and Gracias et al. [15] in the field of smart cities have identified cybersecurity as a key factor for the implementation of a smart city. Furthermore, studies in the area have sought to characterize the various security risks considering theoretical models for identifying vulnerabilities [16–20] and empirical scenarios considering several smart cities such as Dubai, Barcelona, Shanghai, and Como [21–24]. The main research gap is the difficulty of an integrative overview of the main security risks considering a multidisciplinary perspective that can complement the vision of scientific and industrial partners. Combining scientific and industrial partners in the development of smart cities is of paramount importance considering that smart city challenges are complex and multifaceted [25,26]. Therefore, scientific partners can provide in-depth insights into these challenges, while industrial partners can contribute their practical experience and knowledge of market dynamics. By combining these perspectives, smart city projects can be designed and executed in a more holistic and comprehensive manner. In view of this, this study seeks to use a comprehensive perspective that results from the analysis of the role of cybersecurity in smart cities considering several projects, in which several European universities, research centers, and companies participate. The data are extracted from the EU-funded projects database made available by the CORDIS portal. Three research questions (RQs) are posed: RQ1 intends to characterize the dimensions of security risks that can be found in smart cities; then, RQ2 aims to identify measures proposed under these projects to mitigate previously identified security risks; finally, RQ3 seeks to explore emerging security risks that may arise in the face of recent technological and social developments.

This study is organized into the following sections: In the first phase, a theoretical review is made on the topics of information security and security risks in smart cities. After that, the methodological approach and methods applied in the development of the study are presented. This is followed by the presentation and discussion of the results given the research questions previously formulated. Finally, the main conclusions of the study are listed. This last section also sets out the limitations of the study and provides suggestions for future work.

## 2. Literature Review

### 2.1. Information Security

In recent years, the potential and actual impacts of security threats in cyberspace have become evident due to several incidents with direct repercussions on the security of countries and citizens [27–29]. Digital technologies and cyberspace are not only resources and space where one can do better and more efficiently what was once difficult, time-consuming, and costly to do. As argued by Lippert and Cloutier [30], this new reality has created the potential to significantly alter the influence of different groups or actors in the international and national, social, or business scenes.

Information systems, digital communication, and the surrounding digital environment are key elements for successful decision-making processes that aim to produce informed

decisions on critical and sensitive issues. Today, we are able to create, update and store more information than we have ever been able to in the past, but never has that information been as threatened as it is today. To maintain sustainable and competitive infrastructures, which are vital to the survival of nations around the world, there has been a drive to invest in mechanisms and processes that derive from the need to make every possible effort to ensure secure digital resources. Authors such as Knell [31] and Pereira et al. [32] emphasize that the world is highly interconnected, with resources interconnected with structures and networks on a global scale. Therefore, one of the main concerns to ensure a secure protection and safeguarding of information has been the security of information systems, public and private, and their information, which is essential to support the activities of organizations. It also emerges that information security is also vital to building and maintaining customer trust. Consumers are increasingly concerned about the privacy and protection of their personal data. Bleier et al. [33] argue that by demonstrating a serious commitment to information security, organizations can gain customer trust and establish lasting relationships. This is also an important element in the context of smart city adoption, as recognized in this study, in which the relationships established also manifest themselves in the inclusion of new services in smart cities.

Information security addresses multiple domains. Broadly we can look at information security as the protection of data and information of individuals, organizations, and governments from unauthorized access, misuse, disclosure, alteration, or destruction [34]. The importance of information security can be understood in different aspects. Information security must ensure the confidentiality of data, preventing sensitive information from falling into the wrong hands. This is especially critical when it comes to personal data, financial information, trade secrets, or government information. Based in a literature review, Rath and Kumar [35] have concluded that a breach of confidentiality can lead to serious consequences such as identity theft, financial loss, or reputational damage. Information security also aims to ensure data integrity (i.e., its accuracy, completeness, and consistency over time). It is essential that information is not altered in an unauthorized or accidental way, ensuring that it remains reliable and accurate. A lack of data integrity can lead to wrong decisions, errors in processes, and loss of confidence in the information [36]. Finally, information security must ensure that data and systems are available when needed. This implies protecting information against technical failures, cyber-attacks, or natural disasters that may compromise data availability. In their work exploring the relationship between availability and other security issues, Qadir and Quadri [37] have realized that a lack of availability can cause business disruptions, lost productivity, and financial losses. In the context of smart cities, this study also confirms that a lack of availability can have profound and far-reaching impacts on various aspects of urban life. Smart cities rely on a network of interconnected devices, sensors, and systems to deliver critical services and real-time data. Service disruptions can lead to inconvenience for citizens, economic losses for businesses, and even public safety concerns.

Information security, data science, and technology are interconnected domains. Data science involves the extraction of knowledge and insights from data. It encompasses a wide range of techniques, including data collection, data cleaning, data analysis, data visualization, machine learning, and statistical modeling [38,39]. Technology serves as the foundation for both information security and data science. Technological advancements have enabled the rapid growth of digital data and the development of sophisticated tools and algorithms to process and analyze it. Key technological components that support information security and data science include big data platforms, cloud computing, artificial intelligence, machine learning, cryptography, and networking and communication. Hazim and Khan [40] provide several examples of the use of the cloud in organizations and pointed out that cloud services have revolutionized data storage, accessibility, and processing. They provide scalable infrastructure to accommodate large datasets and host data science applications and security solutions. Cloud service providers maintain vast data centers with a wide array of servers, storage, and networking resources. They can allocate and

reallocate these resources dynamically based on demand. According to Brataas et al. [41], this allows users to scale up (add more resources) or scale down (remove excess resources) as needed, without having to invest in physical hardware themselves. Furthermore, cloud platforms offer elasticity, which means they can automatically adjust resources in response to varying workloads. During periods of high demand, the cloud can rapidly provision additional resources, and during low-demand periods, it can release unneeded resources. Biswas et al. [42] state that dynamic scaling ensures efficient resource utilization and cost-effectiveness. Cloud computing and machine learning are two powerful technologies that can be effectively combined to leverage their respective strengths. The integration of machine learning with cloud computing enables scalable and flexible solutions for processing and analyzing large datasets, building sophisticated models, and deploying AI-powered applications [43]. Machine learning tasks often require significant computational resources, especially for training complex models. Peng et al. [44] provide a vision regarding the parallel computing programming mode in the context of cloud computing and argue that it has enabled distributed computing, which can significantly speed up the training process by distributing the workload across multiple servers or nodes. This parallel processing capability can be relevant when training complex models on large datasets.

### 2.2. Security Risks in Smart Cities

ICT plays a key role in the development of smart cities, which combine infrastructure, architecture, objects, and people to improve processes and address social, economic, and environmental problems. Several technologies such as cloud, big data, artificial intelligence, blockchain, and IoT are found in smart cities. Nastjuk et al. [45] advocate that the use of emerging technologies is an enabler of innovation in the context of smart cities, and that it is not limited to the technological perspective of cities, but enables the creation of a smart environment, smart governance, and smart economy. Given the above, it can be inferred that there is no smart city without technology and innovation, as these are the factors that differentiate it from an ordinary city [46]. Necessarily, the adoption of new technologies and the high interconnectivity between them and humans makes smart cities vulnerable to various cyber threats [47,48]. Therefore, protecting the infrastructure, systems, and data from malicious activities is essential to ensure the security, privacy, and reliability of smart city services. Accordingly, there is a need to explore and know mitigation strategies that can address these challenges.

It is important to highlight that the security and privacy challenges of smart cities are not new and that many of them already exist in the isolated use of each of their technologies, but that they now assume a greater impact in the interconnected context of smart cities. The infrastructure of a smart city is composed of thousands of devices and applications that aim to improve processes and bring benefits to citizens. However, the use of these applications and systems can bring several problems related to security and privacy. Elliott and Soifer [49] and Federspiel et al. [50] refer to the vulnerabilities that occur by adopting smart systems based on artificial intelligence, as they not only collect a wide variety of sensitive information from people and their social circles, but also control city facilities and influence citizens' lives. The empirical work carried out also shows that the European projects included in this study consider the challenges inherent in big data, data analytics, artificial intelligence, and machine learning.

Security is seen as a dynamic concept, not a stagnant one, in which we seek to prevent harm by digital and physical means, both direct and indirect [51]. In a smart city, security is looked at in the general component by covering all the features of the city, but it is also included in all the aspects that make it up. Studies by Ghazizadeh et al. [52] use an extended version of the Technology Acceptance Model (TAM) to demonstrate that security is a key factor in technology acceptance. Thus, safety encompasses more than just technical factors, having a strong human-dependent aspect, also including subjective factors related to the perception of individuals [53]. Consequently, the existence of objective and subjective

dimensions of security is assumed. The role of human behavior is also widely highlighted in this study as a dimension is identified.

Cybersecurity plays a key role in protecting critical infrastructure, which includes systems and assets that are essential to the functioning of society and the economy. This infrastructure can include power grids, transportation networks, water treatment facilities, communication systems, and more. Protecting these vital components from cyber threats is critical to prevent the disruptions, unauthorized access, or sabotage that can affect citywide operations [54]. The privacy by design approach advocated in studies such as Drev and Delak's [55] and Romanou's [56] is essential in smart cities that must be supported by a secure architecture and design. Therefore, cybersecurity considerations should be integrated into the architecture and design of critical infrastructure systems from the outset. This involves following the best security practices, conducting risk assessments, and implementing appropriate security controls [57,58].

Protecting data privacy is another area of concern in smart cities' environments. Smart cities generate massive amounts of data from sensors, surveillance systems, and connected devices. These data often contain sensitive information about individuals, including their locations, behaviors, and personal preferences. Data encryption is a fundamental technique used to secure data in transit and at rest. In smart cities, sensitive data such as personal information, financial records, and surveillance footage should be encrypted to prevent unauthorized access or interception by malicious actors [59]. The importance of data minimization and anonymization is associated with the appearance of data encryption. Daoudagh et al. [60] recommend that smart cities should practice data minimization, collecting only the necessary data to fulfill their functions and reducing the risk associated with storing excessive personal information. Furthermore, smart cities must implement robust access controls to restrict access to sensitive systems and data. This involves implementing secure authentication mechanisms such as multi-factor authentication and role-based access control (RBAC) to ensure that only authorized individuals can access specific data [61]. Additionally, smart cities must incorporate secure communication mechanisms. Communication between devices and systems within a smart city's infrastructure should be secured to prevent eavesdropping or tampering. In the review work carried out by Rahouti et al. [62] it is highlighted that the adoption of secure protocols like Transport Layer Security (TLS) and Virtual Private Networks (VPNs) can ensure encrypted and authenticated communication channels. These proposals are relevant to the approach of our study, which identifies cybersecurity challenges related to network vulnerabilities related to unauthorized access and interception of communications.

IoT devices are the backbone of smart cities, enabling connectivity and data exchange among various systems and devices. However, these devices are often vulnerable to cyberattacks due to their limited security measures [63]. Consequently, robust cybersecurity practices are needed to secure IoT devices, including implementing secure authentication, encryption, and regular software updates to prevent unauthorized access or control. Several approaches are proposed in the literature to increase the security of IoT devices. Implementing robust authentication mechanisms such as multi-factor authentication (MFA) is proposed by Ometov et al. [64] to ensure that only authorized users or devices can access the IoT devices. This helps prevent unauthorized access and protects against brute-force attacks. The adoption of secure communication protocols such as Transport Layer Security (TLS) or Secure Shell (SSH) to encrypt data transmitted between IoT devices and the backend systems is proposed by Paul et al. [65]. This prevents the eavesdropping of and tampering with sensitive information. Regular firmware updates are recommended by Gong et al. [66] to keep the firmware of IoT devices up to date by applying regular security patches and updates provided by the manufacturers. This helps address vulnerabilities and ensures that devices are protected against known security risks. Finally, Prazeres et al. [67] employ an approach using test data from different datasets to suggest the implementation of network segmentation to isolate IoT devices from other critical infrastructure systems. This way, even if one device is compromised, it will not provide direct access to the entire

network, reducing the potential impact of an attack. Our study confirms that IoT security plays a critical role in safeguarding not only the devices and systems connected to the internet, but also the entire digital infrastructure and data ecosystem.

Effective cybersecurity in smart cities requires collaboration among various stakeholders, which may include government authorities, urban planners, industry partners, community and citizen groups, and academic and research institutions, among others. As it is advocated by Clement et al. [68], the sharing of information, best practices, and establishing partnerships can help develop comprehensive cybersecurity strategies and responses to emerging threats. Public awareness and education are another pillar of cybersecurity in smart cities. At this level, Williamson [69] states that citizens should be informed about potential risks, advised on secure practices, and encouraged to report any suspicious activities. It is advocated that building a culture of cybersecurity awareness can help prevent attacks and ensure the collective security of the smart city ecosystem. Our study demonstrates that collaborative efforts made at Europe-wide research projects facilitate the sharing of threat intelligence and cybersecurity information among different entities. This allows for a more comprehensive understanding of emerging threats, attack patterns, and vulnerabilities. By staying informed about the latest threats, smart cities can proactively implement countermeasures. Furthermore, it is expected that cybersecurity challenges in smart cities will continue to evolve. Collaborative efforts enable ongoing adaptation to new threats and technologies, ensuring that cities can maintain a robust cybersecurity posture over the long term.

## 3. Methodology

This study adopts qualitative methodology to understand and characterize the role of cybersecurity in smart cities. According to Hammarberg et al. [70], cybersecurity is mostly employed in situations where one seeks to explore and describe in depth the nature, meanings, and characteristics of a particular context, rather than seeking statistical generalizations. In this study, qualitative methodology is used as a tool in the exploratory research of cybersecurity in smart cities. Indeed, qualitative methodology is often used at the beginning of a research process to explore a little-known problem or area of study. It helps to generate hypotheses, identify relevant variables, and gain a deeper understanding of the topic before developing more structured research [71]. This exploratory research of the topic is later complemented by an in-depth survey, which allows exploring the perspectives and experiences of the various research projects in depth, which allows capturing the complexity and context of how cybersecurity is addressed in smart cities' research projects.

The CORDIS database is used to identify projects in smart cities that address cybersecurity risks. CORDIS is maintained by the European Commission to provide information on the objectives, development status, and technical and scientific outcomes of EU-funded research and development projects [72]. The search in this database considered only the repository of projects funded under the Horizon Europe Framework Program, which is the successor to the Horizon 2020 program and spans the period from 2021 to 2027. The program aims to support scientific research, technological development, and innovation across various disciplines and sectors. It is important to highlight that Horizon Europe encourages interdisciplinary research, international cooperation, and open science principles. External and internal validity procedures have been considered. First, we have used consistent search terms to identify relevant projects. Initially, a search was made considering all projects funded by Horizon Europe in the smart cities field. This resulted in the identification of 180 projects. After that, the combination of the three words "smart" and "city" and "security" was applied. It was also ensured that the current state of development of the projects is relatively similar. Therefore, all projects in the approval phase, or approved but not yet started, have been excluded. In total, and after applying the listed restrictions, a total of 62 projects were identified as presented in Table 1. Associated with each project is the date of its implementation (i.e., start date and end date) and a brief

description of its main goal. The identification of the goal of each project was manually carried out considering the main objectives of each project, since the information available in the CORDIS portal is more extensive.

**Table 1.** Characterization of the projects in the sample.

| Project | Start Date | End Date | Goal |
|---|---|---|---|
| 3P-Tec | 1 July 2022 | 31 October 2025 | Evolving three-parent technology to improve crops and seed varieties. |
| ACHILLEUS | 1 November 2022 | 31 October 2025 | Develop a new phenotypic drug discovery platform for targeting cancer stem cell signaling. |
| ALLEGRO | 1 January 2023 | 30 June 2026 | Turning the future innovative, safe and low-energy optical networks. |
| ARTRIGEL | 1 March 2022 | 29 February 2024 | Provide a new innovative injectable treatment for osteoarthritis. |
| ASCEND | 1 January 2023 | 31 December 2027 | Building cities healthier and climate neutral. |
| B-BRIGHTER | 1 October 2022 | 31 March 2025 | A novel top-down lithography approach to bioprinting will overcome current challenges. |
| BoSS | 1 January 2023 | 31 December 2025 | Demonstrate and archive solutions for climate neutrality with a particular focus on coastal cities as an interface to healthy seas, ocean, and water bodies. |
| CLIMABOROUGH | 1 January 2023 | 31 December 2026 | Supporting cities to take up the carbon-neutral challenge. |
| CODECO | 1 January 2023 | 31 December 2025 | Privacy-preserving and innovative edge-cloud management framework. |
| CONCERTO | 1 January 2023 | 31 December 2026 | Safer and greener skies for EU air travel. |
| CONNECTINGHEALTH | 15 June 2022 | 14 June 2024 | Promoting innovative ecosystems in digital health. |
| CULTUURCAMPUS | 1 October 2022 | 31 December 2025 | Shaping a better future for Rotterdam South. |
| DECICE | 1 December 2022 | 30 November 2025 | An open and portable management framework with smart scheduling. |
| DESIRE | 1 October 2022 | 30 September 2024 | Creation of an alternative way forward for the built environment supporting the EU mission of "100 climate-neutral and smart cities". |
| DiaDEM | 1 May 2022 | 30 April 2025 | Provide a digital discovery platform for organic electronics materials. |
| E.T.PACK-F | 1 September 2022 | 28 February 2025 | Provide a ready-to-fly deorbit device based on electrodynamic tether technology. |
| Ebeam4therapy | 1 August 2022 | 31 July 2025 | A compact device for effective delivery of radiotherapy. |
| EDGELESS | 1 January 2023 | 31 December 2025 | Developing groundbreaking serverless computing technology. |
| EHHUR | 1 October 2022 | 30 September 2025 | A sustainable, inclusive, and aesthetically beautiful urban revolution. |
| ENGINEER | 1 October 2022 | 30 September 2025 | Provide a cultural heritage at the heart of development in Cyprus. |
| EV4EU | 1 June 2022 | 30 November 2025 | Vehicle-to-everything strategies facilitate mass uptake of electric vehicles. |
| EXTRACT | 1 January 2023 | 31 December 2025 | Optimization and improvement of extreme data mining. |



**Table 1.** *Cont.*

| Project | Start Date | End Date | Goal |
|---|---|---|---|
| FIBREX | 1 August 2022 | 31 May 2024 | RNA therapeutics for prevention of heart failure. |
| FOR-FREIGHT | 1 September 2022 | 31 December 2025 | Green, cost-efficient, and adaptable multimodal freight transport. |
| GA-VAX | 1 June 2022 | 31 January 2025 | A vaccine against Amyotrophic lateral sclerosis (ALS). |
| GoodMobility | 1 January 2023 | 31 December 2027 | A new logistics approach to urban goods mobility. |
| HYPERIA | 1 September 2022 | 31 August 2024 | High-sensitivity camera sees across the visible and infrared light spectrum. |
| INCYPROnext | 1 June 2022 | 31 May 2025 | Innovative protein stabilization solution for biotechnological and biomedical applications. |
| INSPIRE | 1 April 2022 | 31 March 2024 | En route to the first programmable integrated photonics circuits. |
| L2D2 | 1 October 2022 | 30 September 2025 | Optimizing and standardizing 2D material growth and wafer-scale integration. |
| LEAF | 1 March 2022 | 31 October 2023 | Making science accessible and fun for everyone. |
| LUCERO-BIO | 1 January 2023 | 31 December 2025 | Innovative 3D cell-culture handling platform for preclinical drug screening. |
| MobiSpaces | 1 September 2022 | 31 August 2025 | Moving towards mobility-optimized data governance. |
| MIRACLE | 1 May 2022 | 30 April 2025 | Novel diagnostic platform for virtual biopsy. |
| MITI | 1 April 2022 | 30 September 2024 | Non-ionizing metabolic imaging for predicting the effect of and guiding therapeutic interventions. |
| MLSysOps | 1 January 2023 | 31 December 2025 | Pushing the technological boundaries of autonomic systems through AI/ML. |
| MoSS | 1 May 2022 | 30 April 2024 | Develop an automated DNA storage system based on a novel enzymatic technique for the high-throughput synthesis of DNA. |
| NanoVision | 1 June 2022 | 31 May 2025 | Innovative super-resolution optical microscopy platform. |
| NEBourhoods | 1 October 2022 | 31 March 2025 | Making the Green Deal beneficial for all in Neuperlach. |
| NEB-STAR | 1 October 2022 | 30 September 2025 | New European Bauhaus for green urban transformation. |
| NEMILIES | 1 June 2022 | 31 May 2024 | Provide a room-temperature high-sensitivity infrared detector. |
| NEUTRALPATH | 1 January 2023 | 31 December 2027 | Positive clean energy districts to tackle climate change. |
| NEXUS | 1 June 2022 | 31 May 2025 | Extracellular vesicles as a source for liquid biopsy assay. |
| PANTHEON | 1 January 2023 | 31 December 2025 | Smart city digital twin platform for improved disaster management. |
| PCAVISION | 1 April 2022 | 31 March 2024 | Modern ultrasound imaging for comprehensive prostate cancer diagnostics. |
| PRe-ART-2T | 1 June 2022 | 31 May 2025 | PRe-ART-2T intends to replace low-quality, commercial animal-derived reagent mAbs with high-performing synthetic alternatives. |
| PRISMA | 1 July 2022 | 30 June 2025 | Treating diabetes with a thin-film micropump. |
| PureSurf | 1 May 2022 | 30 April 2025 | Bio-based surfactants from renewable waste streams for the circular economy. |

**Table 1.** *Cont.*

| Project | Start Date | End Date | Goal |
|---|---|---|---|
| Re-Value | 1 January 2023 | 31 December 2026 | Development of climate-neutral solutions for waterfront cities. |
| REWIRE | 1 October 2022 | 30 September 2025 | New framework for around-the-clock data protection. |
| SELFY | 1 June 2022 | 31 May 2025 | Toolbox for more secure, robust, and resilient connected vehicles. |
| SHINTO | 1 October 2022 | 30 September 2025 | Soft, self-healing recyclable materials to support intelligent robotics applications. |
| SiMulTox | 1 April 2022 | 31 March 2024 | Novel platform for multiparametric assessment of drug-related cardiotoxicity. |
| SISHOT | 1 September 2022 | 31 August 2025 | Development of an immediate, real-time single-shot ultrashort laser pulse characterization. |
| SPECTRUM | 1 May 2022 | 30 April 2025 | Innovative switch could solve quantum computer cable problem. |
| SPINE | 1 January 2023 | 31 December 2026 | Looking for smart, green, and inclusive public transport solutions. |
| TAONas-LUAD | 1 April 2022 | 31 December 2025 | Antisense oligonucleotide technology in the treatment of lung adenocarcinoma. |
| TREASoURcE | 1 June 2022 | 31 May 2026 | Turning trash into treasure through a circular economy approach. |
| TrialsNet | 1 January 2023 | 31 December 2025 | Deploy full large-scale trials to implement a heterogenous and comprehensive set of innovative 6G applications. |
| TWIN2EXPAND | 1 January 2023 | 31 December 2025 | Urban planning and design to achieve the EU Green Deal. |
| UP2030 | 1 January 2023 | 31 December 2025 | Urban planning with a focus on climate change. |
| VRP | 1 June 2022 | 31 May 2024 | Robot programming technology that enables automation of high-mix, low-volume production. |

The description and objectives of the projects were subsequently uploaded into NVivo v12.2 software. NVivo is a software program developed for qualitative data analysis that provides a set of tools and features to organize, analyze, and interpret qualitative data. It supports a wide range of qualitative research methods, including content analysis, grounded theory, and thematic analysis, among others. In addition, the software offers advanced analytic tools, including matrix coding, concept mapping, sentiment analysis, and text search queries, to help researchers identify patterns, trends, and relationships in their data. NVivo also offers visualizations such as graphs, diagrams, and word clouds to help interpret and present the results [73]. This study adopted thematic analysis to find the main themes associated with the projects and subsequently identified innovative approaches adopted in the projects to address cyber security risks. Then, a conceptual map of the relationship between the main cybersecurity challenges in smart cities was developed, which guided the discussion of the innovative proposals presented by these projects. Table 2 provides a global vision of the several steps of thematic analysis. Eight phases are considered which include the preparation of data, its coding, the process of identifying and refining themes throughout the process, and the production of the final report.

**Table 2.** Thematic analysis phases.

| Phase | Description |
| --- | --- |
| Data preparation | The data collected on the CORDIS platform for each project are compiled into an individual report. Information on project name, implementation dates, summary, objectives, and technical description are extracted. These elements are available in "fact sheet" and "reporting" sections on CORDIS platform. A script written in Python was developed to automatically collect this information. |
| Codification | NVivo v12.2 software is used to assign codes to text segments or significant units of data that represent a specific theme. |
| Identification of initial themes | Initial emerging themes are identified and highlighted by Nvivo v12.2 software. Themes represent groups of codes that share similar characteristics. |
| Review of themes | This step aims to review the identified themes to ensure that they are aligned in the cybersecurity domain and are appropriately relevant and comprehensive. It is also relevant to ensure that the themes correctly capture the perspectives of each project. |
| Definition and naming of themes | Each theme is associated with a descriptive name to assist in the analysis and production of the final report. |
| Analysis of themes | The themes identified in the previous step are analyzed to understand the relationships between them and to highlight how they contribute to the research questions formulated in the study. |
| Interpretation of results | The results obtained are discussed considering the research objectives and the literature in the field. Implications of the themes are noted to guide the discussion of the findings. |
| Reporting of results | A detailed report of the identified themes and their definitions is produced. Original citations are also collected from the studies that support each theme. The implications of the research are also noted. |

## 4. Results

The thematic analysis began by exploring the most frequent terms associated with the objectives and description of each project. Table 3 summarizes the top 10 frequent themes considering the total number of times is the theme was identified in projects. The most frequent themes include "smart cities" with 96 occurrences, "data-driven" with 74 occurrences, "artificial intelligence" with 53 occurrences, "machine learning" with 44 occurrences, and "digital environments" with 41 occurrences. Other themes that stand out with less frequency are "privacy", "sensors", "education", "sustainability", and "collaborative environment". A significant portion of the terms identified are not directly related to cybersecurity in its multiple aspects, nor to the area of application to smart cities, but emerge as important elements in the methodological and organizational processes of each project, such as the search for artificial intelligence support in data analytics for decision making and the role of collaborative environments in the development of these technological solutions. Some common architectural components for the development of smart cities are also identified, such as the exploitation of information generated through data analytics techniques, the evolution of artificial intelligence that enables a more personalized interaction with citizens, information privacy, and the role that the IoT can assume in various areas, such as urban mobility, energy efficiency, and sustainability.

The next step in presenting the results was to create a cognitive map to capture the various dimensions of cybersecurity risks in smart cities. Figure 1 presents the multiple dimensions captured in a cognitive map. A total of 7 dimensions and 31 sub-dimensions were identified. Cybersecurity risks in smart cities are grouped under dimensions such as "infrastructure vulnerabilities", "data privacy", "network vulnerabilities", "access control", "IoT devices", "security standards and regulation", and "human behavior". It is noteworthy that there is a strong dependence between the various dimensions. Infrastructure vulnerabilities are divided into areas such as sensors and networks, which then give rise to vulnerabilities supporting network communication in smart cities and IoT devices. It is also highlighted that the modular and collaborative structure in the development of smart cities with multiple manufacturers and technologies makes it difficult to manage the entire network in an integrated way. Finally, human behavior makes the emergence of social engineering attacks a possibility, in addition to being a challenge for digital inclusivity.

Finally, the security risks identified above are addressed by European research projects. Several solutions are proposed in the technological, organizational, and human behavior domains. Table 4 identifies mitigation proposals put forward by the research projects. Three

quotes from the projects are provided, extracted from NVivo v12.2 software, on how the research projects address mitigation proposals in each dimension. It is highlighted that the mitigation proposals presented by the research projects highlight their scope of application as smart infrastructure, urban mobility, energy and sustainability, and quality of life. It also stands out that the same project (e.g., EDGELESS, PANTHEON) addresses multiple dimensions of cybersecurity in a smart city.

**Table 3.** Top 10 most frequent themes.

| Final Theme | Associated Terms | Absolute Frequency | Relative Frequency |
|---|---|---|---|
| Smart cities | Intelligent cities, digital cities, future cities | 96 | 0.1605 |
| Data-driven | Big data, data analytics, data-based, data-centric | 74 | 0.1237 |
| Artificial intelligence | Intelligence, AI-driven systems, intelligent agents, neural networks | 53 | 0.0886 |
| Machine learning | Machine, automation, data mining | 44 | 0.0736 |
| Digital environments | Virtual environments, human-center, behavior, cyberspace | 41 | 0.0686 |
| Privacy | Secure, confidentiality, cyber-attack, private data | 40 | 0.0669 |
| Sensors | IoT, network, monitoring | 36 | 0.0602 |
| Education | Learning, training | 33 | 0.0552 |
| Sustainability | Sustainable, green practices, circular economy, renewable | 21 | 0.0351 |
| Collaborative environment | Collaboration, empowerment, co-creating, stakeholders, community, co-design | 17 | 0.0284 |

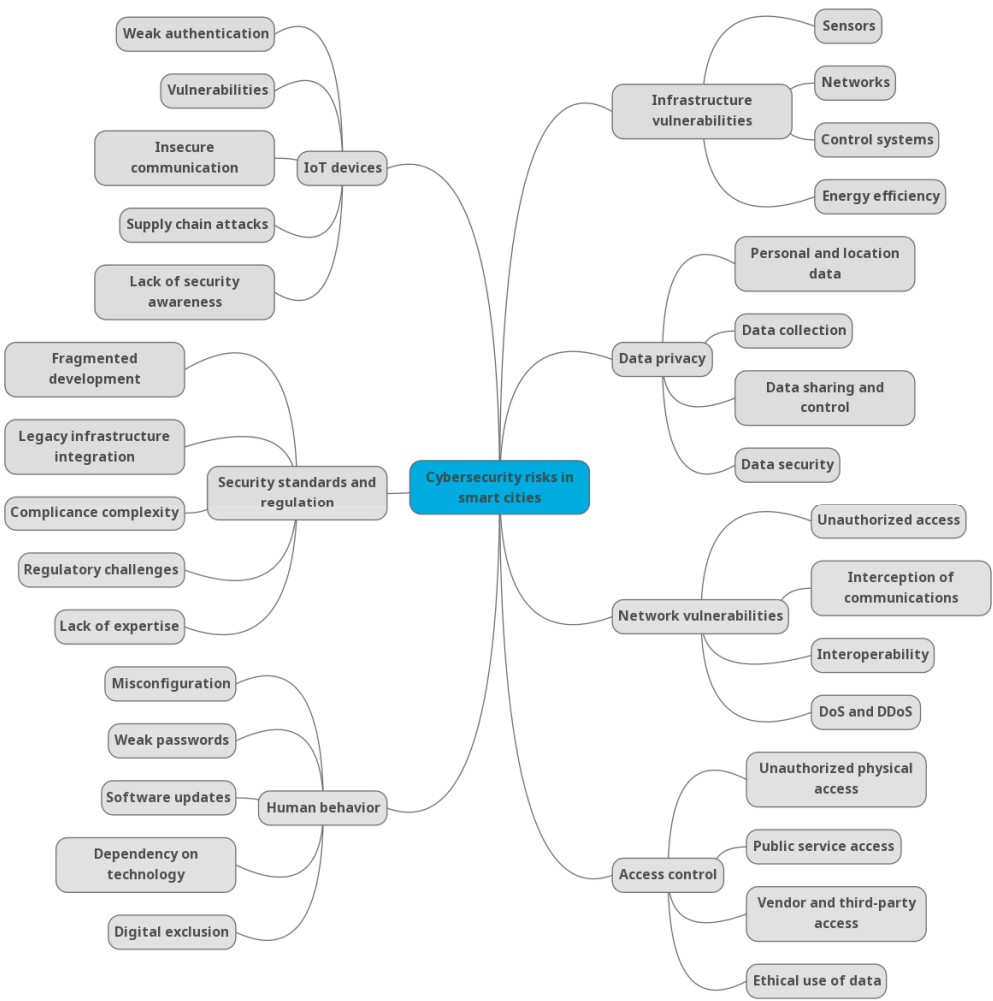

**Figure 1.** Cognitive map of cybersecurity risks.

**Table 4.** Mitigation proposals for cybersecurity risks.

| Dimension | Mitigation Action | Project Vision |
|---|---|---|
| Infrastructure vulnerabilities | Vulnerability assessment<br>Risk management plan<br>Ethical hacking<br><br>Continuous monitoring | "…promote the conduct of comprehensive surveys on multi-hazard disaster risks and the development of regional disaster risk assessments and maps" (PANTHEON)<br>"Solutions include innovative urban design, behavioral nudging, smart technological and data-driven solutions to reduce actual and perceived road safety risks…" (REALLOCATE)<br>"Co-creating co-benefits for the neighborhood and city through multi-functional use of spaces and infrastructures." (NEB-STAR) |
| Data privacy | Privacy by design<br>Data minimization<br>Transparent data practices<br><br>User control and rights | "The SPINE approach involves the creation of (a) innovative simulation and Digital Twining (DT) tools, along with open data and behavioral models, that will allow the building of scenarios combining different mobility interventions…" (SPINE)<br>"…data sharing, advanced processing for detection of malicious events and decision-making…" (SELFY)<br>"Specifically, the platform will feature enhanced data infrastructures and AI and big-data frameworks, novel data-driven orchestration and distributed monitoring mechanisms, a unified continuum abstraction and cybersecurity and digital privacy across all software layers." (EXTRACT) |
| Network vulnerabilities | Risk assessment<br>Prevention systems | "…resilience, increased ability to adapt and respond to cyber-threats and cyber-attacks…" (SELFY)<br>"…a scalable AI/ML assisted control and orchestration system, responsible for autonomous networking, dynamic and constrained service provisioning…" (ALLEGRO)<br>"This ambitious challenge will be met via distributed computing solutions to partition the edge environment in clusters, each managed as a local decentralized serverless platform." (EDGELESS) |
| Access control | Intrusion detection<br>Authentication and authorization mechanisms<br>Secure device management<br><br>Role-based access controls | "Clusters will cooperate with each other and with all the layers in the edge-cloud continuum to compose complex applications on-demand through a FaaS paradigm." (EDGELESS)<br>"MLSysOps will employ a hierarchical agent-based AI architecture to interface with the underlying resource management and application deployment/orchestration mechanisms of the continuum." (MLSysOps)<br>"…the camera will be able to capture visible features, key SWIR ?fingerprints? for physiochemical analysis and fluorescence signals from non-visible features." (HYPERIA) |
| IoT devices | Secure access controls and protocols<br>Secure communications protocols<br><br><br>Security monitoring | "…platform and technologies will be combined with IoT infrastructure, multi-source data (satellite and in situ data, social networks, historical data) to create a tool for assessment of risks, vulnerability and capacity assessment." (PANTHEON)<br>"The proposed scalable and multifunctional cybersecurity platform will ensure the security throughout the life of the IoT devices with continuous security auditing, trust computing and theorem proofs for defining an hw-based microarchitecture for enhanced protection targeting to open-hardware/software vulnerabilities." (REWIRE)<br>"…we developed an uncooled IR sensor prototype based on a nanoelectromechanical system (NEMS), called NEMILIE, which can reach unprecedented sensitivity at room temperature." (NEMILIES) |

**Table 4.** *Cont*

| Dimension | Mitigation Action | Project Vision |
|---|---|---|
| Security standards and regulation | Adoption of international standards<br>Smart grid security standards<br><br>Articulate with local and national frameworks | "The project develops the 5UP methodological framework that supports cities in (i) UP-dating those policies, codes, regulations that need to be left behind to make room for the new vision." (UP2030)<br>"REWIRE envisions a holistic framework for continuous security assessment of open-source and open-specification hardware and software for IoT devices and the development of cybersecurity certification in accordance with the requirements and guidelines of recent EU regulation Cyber security Act3." (REWIRE)<br>"...compliance applicable to the three "thrusts" of Clean Aviation and a first status of comprehensive digital framework of formalized collaborative tooled and model/simulation-based processes for certification." (CONCERTO) |
| Human behavior | Education and awareness<br>Ethical guidelines<br>Behavioral analytics and predictive models<br>Collaboration and partnerships<br><br>Community engagement | "By this, we will set the foundation for a school of thought and practice, and establish a scaling framework for widespread learning across the EU utilizing digital infrastructure, stakeholder involvement and empowerment across a partner community of European cities, youth organizations, NGOs, academia, etc." (DESIRE)<br>"Combining co-creation and entrepreneurship, putting culture and creativity at the core of the transformation process, the project will deliver accessible and empowering solutions to make the EU-Green Deal beneficial for all in NPL and beyond" (NEBourhoods)<br>"TWIN2EXPAND embraces the international networking ethos of the SDGs to achieve scientific excellence in the R&I of the built environment, fully embedding it within the quadruple helix by fostering collaboration with local authorities and other stakeholders." (TWIN2EXPAND) |

## 5. Discussion

Research projects promoted under Horizon Europe are a prime source of information for understanding security risks in smart cities. The EU has encouraged and funded several initiatives to promote the concept of smart cities and bring the poorest regions of the EU closer to the richest. European smart city projects often involve partnerships between cities, businesses, research institutions, and civil society organizations. They provide a framework for collaboration, sharing knowledge and good practices, and provide financial resources for implementing smart city solutions. The exploration of these initiatives helps us to answer RQ1 and allows us to synthesize cyber security risks in seven dimensions, and it is noted that the projects cover all these multiple areas. The role of these initiatives in providing a comprehensive view of security risks is also noted. Thus, the risks of using emerging technologies are compounded by regulatory difficulties in implementing smart cities and the interaction of humans with smart cities.

The infrastructure of a smart city is a critical element. Mitigation measures, as indicated in RQ2, should address this central point of smart city architecture. As cities become more reliant on technology to operate, critical infrastructure becomes an attractive target for cyberattacks. Furthermore, the interconnectivity of devices and systems in smart cities creates a larger attack surface for potential attackers. Several mitigation measures can be implemented, such as risk assessment and management, continuous monitoring, and intrusion detection, among others. The vulnerabilities existing in the security systems of a smart city exist in several forms, such as security breaches in communication networks, failures in monitoring and control systems, outdated systems, and lack of proper authentication, among others [74,75]. Effectively, the continuous monitoring of security risks is an activity that must be constantly performed to detect and respond to threats in real time. In Herath and Mittal [76], the role of security monitoring tools, data analytics, and artificial intelligence to identify anomalies and suspicious activities is highlighted. Artificial intelligence also poses new risks such as autonomous learning models and Deepfake, which refers to the malicious use of deep learning technology to create highly realistic and often misleading fake content. Moreover, security risk management in smart cities is an ongoing process. Good practices in this area are addressed by Hui [77] and Younus [78], by suggesting conducting regular risk assessments, updating security measures according to emerging threats, and participating in collaborative actions with other smart cities to share knowledge and experiences. Privacy is another key issue in smart cities. The large-scale collection of personal information and constant monitoring of individuals' activities can raise legitimate concerns. Smart cities must be transparent about their data collection practices and involve citizens in the decision-making process. This includes providing clear information about the technologies used, the expected benefits, and the associated privacy risks. Data collection is an activity that should be conducted in a transparent and consenting manner, informing citizens of what data are collected, how they are used, and who has access to it. How data are processed and stored is another point that should deserve attention. Faisal [79] points out that one way to protect citizens' privacy is to ensure that the data collected are minimized, anonymized, and aggregated wherever possible. This means removing personally identifiable information and grouping data at aggregate levels to avoid identifying specific individuals.

The cybersecurity risks identified are also strongly connected with the emergence of data-driven solutions in smart cities. Therefore, answering RQ2 requires considering that Artificial Intelligence (AI) plays a pivotal role in processing and analyzing vast amounts of data to optimize services and infrastructure. AI's rapid evolution has revolutionized various aspects of smart cities, enabling them to make data-driven decisions, automate processes, and enhance citizen experiences. In transportation, AI-powered algorithms manage traffic flow, optimize public transportation routes, and predict maintenance needs for infrastructure [80]. In energy management, AI enables real-time monitoring, demand-response systems, and the optimization of energy consumption [81]. However, as smart cities become more reliant on AI-driven systems, their susceptibility to cyber threats

increases significantly. Effectively, as AI becomes increasingly integrated into smart cities, it also exposes these urban environments to new cybersecurity challenges. Cyber threats pose significant risks, potentially leading to data breaches, service disruptions, privacy violations, and even physical harm [82,83]. One significant concern is the potential for AI models to be manipulated, leading to biased decision-making or malicious actions. As stated by Anthi et al. [84], AI algorithms may be vulnerable to adversarial attacks, where malicious input data are crafted to deceive the model. Additionally, Anwar and Ali [85] reveal that the centralization of data in smart cities presents a potential target for hackers seeking to steal sensitive information or hold it for ransom.

Dealing with the challenges of standards and regulation in smart cities requires a collaborative and adaptive approach. Ensuring interoperability between the different systems and devices in a smart city is a challenge that must be considered in RQ2. Madsen [86] states that adopting open standards allows systems to communicate with each other efficiently, avoiding dependence on specific vendors and promoting healthy competition. In addition, standards also help to ensure data security and privacy. Importantly, smart cities involve a wide range of stakeholders, including governments, businesses, academic institutions, and citizens. Promoting and establishing mechanisms for dialog and collaboration among these stakeholders can be used to develop appropriate standards and regulations. Several approaches can be implemented in this field, such as the use of discussion forums, public consultations, and public–private partnerships [87,88]. Citizens are also a key pillar in the implementation of smart cities. Citizen ethics are important because they influence people's behavior toward the use of these technologies and how they interact with the smart city environment. Ethical citizens use the technologies available in smart cities responsibly. This includes not using digital tools for illegal or harmful activities, respecting laws and regulations related to cybersecurity, and avoiding the misuse of others' personal data. Additionally, citizens' ethics can be promoted through educational programs and awareness about security in smart cities. According to Ziosi [89], this involves disseminating information about the risks and benefits of the technologies used, the importance of data protection, and raising awareness about citizens' rights and responsibilities in a smart city environment.

The strong technological interconnection in smart cities allows them to collect, analyze and share data efficiently, making it possible to operate and manage various urban aspects intelligently. However, the proliferation of new technologies and the need to have a highly integrated environment means that new security risks are continuously emerging. The research projects launched between 2022 and 2027 should not only address the main current security risks, but should identify new risks that may emerge in the next decade, which gives us relevant information to address RQ3. The PANTHEON project presents the following strategic vision:

> "*PANTHEON will design and develop a Community based Digital Ecosystem for Disaster Resilience utilizing Smart City Digital Twin (SCDT) technology and leveraging new and emerging technologies and innovations to improve risk assessment, reduce vulnerability, and building community disaster resilience.*" (PANTHEON)

The SCDT approach proposed in PANTHEON refers to a comprehensive framework that combines digital technologies, data analytics, and modeling techniques to enhance disaster preparedness, response, and recovery in a smart city environment. A SCDT is referred to in Deren et al. [90] as a virtual replica or simulation of a city's physical infrastructure, systems, and processes. A SCDT integrates various data sources, such as Internet of Things (IoT) devices, sensors, and real-time data feeds, to create a dynamic representation of the city. SCDT is a cutting-edge concept that contributes to better infrastructure management, urban planning and design, and real-time decision making. Caprari et al. [91] argue that digital twins enable city planners and architects to test and visualize various urban planning scenarios before implementing them in the physical world. This helps in optimizing the layout of infrastructure, buildings, transportation networks, and public spaces, leading to more efficient and sustainable cities. Furthermore, the availability of real-time data from

an SCDT empowers city officials to make informed and data-driven decisions promptly. Cheng et al. [92] provide the example that during emergencies or unexpected events, such as natural disasters or traffic congestion, authorities can quickly assess the situation and respond effectively.

Another emerging issue where the roles of cybersecurity and sustainability are aligned is represented in the ambition of the CONCERTO project.

> *"Clean Aviation's ambition to go over decisive impactful steps in demonstrated disruptive aircraft performance compatible with 2035 EIS will only be possible if the future regulatory framework is not an impediment to innovation."* (CONCERTO)

As reported by Timperley [93], the aviation industry is a significant contributor to global greenhouse gas emissions, primarily due to the burning of fossil fuels in aircraft engines. Clean aviation aims to address this issue by adopting various strategies and technologies to increase the sustainability of air travel [94–96]. Furthermore, CONCERTO project exposes the potential cybersecurity risks raise in this context. As aviation becomes increasingly digitized and connected, with the adoption of technologies such as autonomous systems, IoT devices, and cloud computing, the potential cybersecurity risks also increase. Furthermore, a sustainable organization aims to be resilient and adaptable to various challenges, including cyber threats. Cybersecurity measures play a significant role in safeguarding an organization's ability to continue sustainable practices in the face of potential disruptions or attacks. Therefore, sustainable practices should be integrated into business continuity and disaster recovery plans. Several strategies like redundancy, geographic diversity, and robust backup systems can help organizations to protect data and infrastructure from cyber-attacks, natural disasters, or other disruptive events [97–99].

Another area addressed by the projects is smart agriculture as in the 3P-Tec project.

> *"Changes in climate extremes (e.g., heatwaves, floods and drought) require new breeding technologies to allow farmers to achieve high yields with climate-robust crops."* (3P-Tec)

Cybersecurity in smart agriculture is a growing concern as the agricultural industry increasingly adopts digital and connected technologies to optimize crop production and management. Smart farming, also known as precision agriculture, involves the use of sensors, drones, IoT, and other technologies to collect data and automate agricultural processes. While these technologies offer significant benefits such as increased efficiency, cost reduction, and higher productivity, they can also expose farming operations to cyber threats. Some of the key cybersecurity challenges in smart agriculture include connectivity, data protection, phishing, and malware [100–102].

Finally, another area highlighted by research projects is the role of entrepreneurship in cybersecurity. This perspective is addressed in projects such as EHHUR.

> *"Future adopters will benefit from a DSS tool, supported by a capacity building programme, that offers the ideal mix of best technical solutions, optimal financing schemes and engagement tools and social innovation practice."* (EHHUR)

DSS tools can be used in the context of smart cities for resource allocation and predictive analytics. Mahmood et al. [103] state that DSS can optimize resource allocation by analyzing data on resource availability and demand, ensuring that resources are utilized optimally. Osman and Elragal [104] add that DSS can employ predictive analytics to forecast future trends and events. This capability will allow city authorities to proactively address potential issues, plan for contingencies, and make informed decisions to improve the city's resilience. Also linked to the role that technology can play is the importance of entrepreneurship in this field. Entrepreneurs have the ability to identify gaps in the cybersecurity market and understand the needs of end users. They can find business opportunities by realizing unsolved problems or inadequate solutions existing in the cybersecurity field. This view is shared by Aiyer et al. [105] and Alspach [106], when highlighting that many startups have emerged in the cybersecurity area, seeking to develop more effective and

affordable solutions. Entrepreneurs found these startups and brought together specialized cybersecurity talent to address emerging challenges.

## 6. Conclusions

Cybersecurity is a key element in the context of smart cities. The high connectivity and high technological dependence of smart cities increase the emergence of cyber risks. The security of digital systems and infrastructures is essential to ensure data protection, citizens' privacy, and the efficient functioning of smart cities. The approach adopted in this study complemented the systematic review works in the identification of key factors in the implementation of smart cities and also the empirical studies carried out in the specific context of smart city implementation. The exploration of emerging security risks and the analysis of proposals to mitigate these risks based on research projects carried out in the European context allowed the risks identified to reflect not only the scientific perspective of the research work, but also the technical and practical solutions that have been proposed to make cities smarter supported by technology and optimization of available resources. This study identified that the main security challenges facing smart cities cover several areas, such as the protection of technological infrastructure, data privacy, network security, access control, IoT device security, the adoption of standards and regulations in the area, and human behavior. All these factors contribute to the implementation of robust cybersecurity measures in all aspects of the planning, development, and operation of smart cities, to ensure a safe and reliable digital environment for all involved. Furthermore, this study identified emerging research areas in projects addressing topics related to SCDT, clean aviation, smart agriculture, and data analytics that bring new security challenges to smart cities.

This study offers both theoretical and practical contributions. In the conceptual dimension, this study contributed to identifying the main cybersecurity risk areas to which smart cities are exposed. It was possible to build a cognitive map consisting of seven dimensions and identify mitigation measures to address these risks. Furthermore, the identification of emerging cybersecurity research areas in smart cities includes the implementation of the concept of digital twins, smart aviation, smart agriculture, and the emergence of new cybersecurity startups that bring new challenges to the management of smart cities using emerging technologies and living test labs. In the practical dimension, the findings of this study are relevant for researchers in the field who want to launch new approaches and practical solutions to mitigate the risks of smart cities and deepen their knowledge on this area. Also, the practical results of this study are relevant for the establishment of public policies that can encourage the development of regulatory instruments that facilitate collaboration between the various stakeholders involved in the planning and implementation of a smart city.

Some limitations should also be highlighted. The database of research projects provided by CORDIS is a rich and complementary instrument of the scientific production associated with each of the areas and allows us to have a more comprehensive view of the projects implemented on a global scale considering the perspective of the various stakeholders. However, the projects included in this database include only projects funded by the European Commission, although they may include partners outside the European Union. It is also relevant to note that the objectives proposed by these projects at the time of their acceptance by the EU often have to be reformulated throughout their implementation phase. This study does not cover this situation by capturing only the objectives of the projects in their design phase. In future work, it would be interesting to explore how the objectives of each project are modified with the emergence of new security risks throughout its implementation cycle. Another limitation of this study is the absence of a quantitative and comparative metric that would allow us to assess the current relevance of each security risk and its potential for growth over the course of this period of learning. The study only captured each of the dimensions of security risks without measuring their relative prevalence. Therefore, for future works, it is suggested that each of the security risks be

assessed considering their practical impact on building a smart city. This can be achieved by carrying out a study considering the specific scenario of a smart city or adopting a more globalizing perspective and reflecting the perception of the various stakeholders (e.g., companies, public institutions, citizens, etc.) about these risks.

**Funding:** This research received no external funding.

**Institutional Review Board Statement:** Not applicable.

**Informed Consent Statement:** Not applicable.

**Data Availability Statement:** Data available on request from the author.

**Conflicts of Interest:** The authors declare no conflict of interest.

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
