# Peer review of "Prospects of Cybersecurity in Smart Cities"

_futureinternet, doi:10.3390/fi15090285_

Round 1

Reviewer 1 Report

The idea to analyze the set of current EU Horizon Europa-based projects is good. It could attract a broad auditory of readers.

On the other hand, the cybersecurity issues are not well-described. General notes and introductory/common sense reasoning [to non-specialists] is frequently used in this research.

In the beginning it is written that the main keywords in this direction should be AI, Machine Learning. Then nothing of this is included in the so-called 'qualitative analysis'. The methods of the 'qualitative analysis' are insufficiently described. As a whole, the analysis part is poor.

Instead, for example, in Table 1 /Section METHODOLOGY we read projects with goals like 'Novel drug-discovery platform to target cancer stem cells'. There is no main idea, the possible 'central line' is hidden between different non-important or general EU direction notes.

As a result, the current goals, problems, and features of IoT / Smart City security ecosystems are hidden or overlooked. The analysis part is rather shallow. It consists of foreign words arranged in a rather poor manner.

As a result, the cognitive map on Fig. 2 is incomplete. For example, it is far from DeepFake attacks, and/or the usage of chatGPT, LAMBDA, and/or other contemporary significant cases. Same for other important issues of IoT and other cybersecurity applications.

The reference part should be enlarged, its analysis should be cardinally improved.

The discussion part, especially what is written in lines 360-404 should be rewritten and cardinally improved.

Conclusions contain too many introductory notes. Please transfer them to the first section of this file, or delete them.

The general descriptions like in line 423 are very frequent examples, please remove them: 'Smart cities are attractive targets for hackers and cybercriminals as they offer a wide attack surface'.

It is written: 'This study offers both theoretical and practical contributions'.

For pity, in this version the experimental part is written in a fuzzy manner, it is simply poor.  Actually, all possible applications are addressed to a future work (line 462). In my humble opinion, this version should be rejected but I want to give the possibility for cardinal improvements of this work. Maybe the practical results, and comparisons of what is depicted in Fig. 2 to the best analogs are overlooked, and will be added during the next attempt.

This version looks like a conference poster or a description of a future project/stage 0. Please improve it.

Style corrections are welcome

Author Response

We appreciate the review suggestions and comments received by the reviewer. These elements are key to improving the final quality of the manuscript. Below we respond to each issue raised.

Review #1

The idea to analyze the set of current EU Horizon Europa-based projects is good. It could attract a broad auditory of readers.

Author’s response: Thank you very much for your global appreciation and improvement suggestions provided in next points.

On the other hand, the cybersecurity issues are not well-described. General notes and introductory/common sense reasoning [to non-specialists] is frequently used in this research.

Author’s response: Thank you for the recommendation and we also consider it is important to use more scientifically precise language in the cybersecurity field, considering the specific technical challenges that can be encountered at the cybersecurity level in smart cities.

In the beginning it is written that the main keywords in this direction should be AI, Machine Learning. Then nothing of this is included in the so-called 'qualitative analysis'. The methods of the 'qualitative analysis' are insufficiently described. As a whole, the analysis part is poor.

Author’s response: Thanks for your pertinent recommendation to better provide a description of the employed methods. We have provided a Table 2 that provides a description of all phases of the thematic analysis. Eight phases are considered which include the preparation of data, its coding, the process of identifying and refining themes throughout the process, and the production of the final report. Furthermore, we have replaced the image that presents the word cloud of terms by a Table 3 that unequivocally counts the most frequent terms considering its absolute and relative frequency. This provides more robust information to the readers.

Instead, for example, in Table 1 /Section METHODOLOGY we read projects with goals like 'Novel drug-discovery platform to target cancer stem cells'. There is no main idea, the possible 'central line' is hidden between different non-important or general EU direction notes.

Author’s response: Thanks for your observation. We have identified this issue and corrected it for all the projects. We also consider that it is important to present the main goals of the project but without giving too much detail that may overextend the size of the table.

As a result, the current goals, problems, and features of IoT / Smart City security ecosystems are hidden or overlooked. The analysis part is rather shallow. It consists of foreign words arranged in a rather poor manner.

Author’s response: We have deleted the initial Table 1 and provide a new Table 3 that presents the most frequent final themes. This table provides information regarding the final themes, associated terms, absolute frequency, and relative frequency.

As a result, the cognitive map on Fig. 2 is incomplete. For example, it is far from DeepFake attacks, and/or the usage of chatGPT, LAMBDA, and/or other contemporary significant cases. Same for other important issues of IoT and other cybersecurity applications.

Author’s response: Thanks for your observation. Currently the old Figure 2 is the new Figure 1. We would like to highlight that the cybersecurity risks identified are the result of the thematic analysis carried out based on the information provided by the projects. Therefore, there are cybersecurity risks that are widely recognized in the literature but that do not stand out in the specific context of smart cities. The cybersecurity risks suggested by the reviewer are relevant and are found within large groups such as "personal and location data", “data security”, “Insecure communication” and “supply chain attacks”. We also consider relevant to highlight their importance in the discussion of the results to allow a more correct and comprehensive interpretation of cybersecurity risks in smart cities.

The reference part should be enlarged, its analysis should be cardinally improved.

Author’s response: We have included new references in our study to better support the relevance of our findings. Currently, this paper has 86 references.

The discussion part, especially what is written in lines 360-404 should be rewritten and cardinally improved.

Author’s response: Thanks for your suggestion. We have reformulated the discussion section. We have improved the discussion of the results to highlight emerging lines of research and their respective scientific relevance. New references have been included, respectively:

Caprari, G.; Castelli, G.; Montuori, M.; Camardelli, M.; Malvezzi, R. Digital Twin for Urban Planning in the Green Deal Era: A State of the Art and Future Perspectives. Sustainability 2022, 14, 1-16. https://doi.org/10.3390/su14106263

Cheng, R.; Hou, L.; Xu, S. A Review of Digital Twin Applications in Civil and Infrastructure Emergency Management. Buildings 2023, 13, 1-29. https://doi.org/10.3390/buildings13051143

Peeters, P.; Higham, J.; Kutzner, D.; Cohen, S.; Gössling, S. Are technology myths stalling aviation climate policy? Transportation Research Part D: Transport and Environment 2016, 44, 30-42. https://doi.org/10.1016/j.trd.2016.02.004

Gössling, S.; Lyle, C. Transition policies for climatically sustainable aviation. Transport Reviews 2021, 41, 643-658. https://doi.org/10.1080/01441647.2021.1938284

Bergero, C.; Gosnell, G.; Gielen, D.; Kang, S.; Bazilian, M.; Davis, S.J. Pathways to net-zero emissions from aviation. Nature Sustainability 2023, 6, 404-414. https://doi.org/10.1038/s41893-022-01046-9

Ting, L.; Khan, M.; Sharma, A.; Ansari, M.D. A secure framework for IoT-based smart climate agriculture system: Toward blockchain and edge computing. J. Intell. Syst. 2022, 31, 221-236. https://doi.org/10.1515/jisys-2022-0012

Zanella, A.R.; Silva, E.; Albini, L.C.P. Security challenges to smart agriculture: Current state, key issues, and future directions. Array 2020, 8, 1-12. https://doi.org/10.1016/j.array.2020.100048

Mahmood, O.A.; Abdellah, A.R.; Muthanna, A.; Koucheryavy, A. Distributed Edge Computing for Resource Allocation in Smart Cities Based on the IoT. Information 2022, 13, 1-12. https://doi.org/10.3390/info13070328

Osman, A.; Elragal, A. Smart Cities and Big Data Analytics: A Data-Driven Decision-Making Use Case. Smart Cities 2021, 4, 286-313. https://doi.org/10.3390/smartcities4010018

Conclusions contain too many introductory notes. Please transfer them to the first section of this file, or delete them.

Author’s response: We have followed your recommendation and deleted these introductory notes that are redundant considering the structure of Introduction section.

The general descriptions like in line 423 are very frequent examples, please remove them: 'Smart cities are attractive targets for hackers and cybercriminals as they offer a wide attack surface'.

Author’s response: We have deleted this sentence.

It is written: 'This study offers both theoretical and practical contributions'.

For pity, in this version the experimental part is written in a fuzzy manner, it is simply poor.  Actually, all possible applications are addressed to a future work (line 462). In my humble opinion, this version should be rejected but I want to give the possibility for cardinal improvements of this work. Maybe the practical results, and comparisons of what is depicted in Fig. 2 to the best analogs are overlooked, and will be added during the next attempt.

Author’s response: Thank you for the opportunity. We believe that the greater methodological rigor in the presentation of the qualitative analysis allows for a more rigorous identification of themes. Also, the discussion of the results has been improved to give a more comprehensive view of security risks. New references were also included in the discussion of the results, which allows to better substantiate the evidence collected and discuss its relevance in the context of smart cities.

This version looks like a conference poster or a description of a future project/stage 0. Please improve it.

Author’s response: We consider that the work of theoretical review and empirical work of data analysis of the projects allows to have a sufficiently comprehensive view for the work to be published in a journal. The number of references used and especially the discussion of the results offers both theoretical and practical contributions that are relevant and that justify the publication of the work. Furthermore, the theme of the paper is perfectly aligned with the scope of the Future Internet journal.

Style corrections are welcome.

Author’s response: Thanks for the suggestion. We have proofread the full manuscript.

Reviewer 2 Report

The paper is interesting and addresses are relevant topic. But the methodology and the conclusions are weak, from my point of view. Please, update you methodology to discuss some critical validity problems:

- Why is your paper based only in European Projects? Is this dataset general enough to ensure your conclusions are not subjective? How did you separated the technical proposal from the political content of this kind of projects?

- Have you analyzed the internal and external validity?

- In CORDIS, the full project proposal is not available. How did you solve this situation? How are you sure the short description are you considering is a good project descrition? Did you employed any text quality indicator to reject those sentence or paragraph with a poor quality?  

Although you methodology is qualitative, for sure you employed techniques to identify synonyms, expressions with an equivalent meaning, etc. This is not described in your paper.

Results and conclusions cannot be considered strong until all this information is provided and can the analyzed. Please, update your paper accordingly. 

Author Response

Response to Reviewer

We appreciate the review suggestions and comments received by the reviewer. These elements are key to improving the final quality of the manuscript. Below we respond to each issue raised.

Review #2

The paper is interesting and addresses are relevant topic. But the methodology and the conclusions are weak, from my point of view. Please, update you methodology to discuss some critical validity problems:

Author’s response: Thanks for your positive evaluation regarding the relevance of this study. We have followed your recommendation and significantly improved the methodology section. We have included a new Table 2 that describes the several phases of the thematic analysis process. Associated with this we also had the reformulated presentation of the frequent terms found in the projects which is now presented in Table 3. Finally, the presentation of the conclusions has been improved with an increased focus on presenting the contributions of the study and the introductory notes previously presented in this section have been eliminated.

- Why is your paper based only in European Projects? Is this dataset general enough to ensure your conclusions are not subjective? How did you separated the technical proposal from the political content of this kind of projects?

Author’s response: Database choice is always a crucial element that limits the scope of the study.  The European Commission has played a key role in supporting and engaging public and private partners in the development and implementation of research and innovation projects. On an international scale there are other entities, but the CORDIS database used is sufficiently comprehensive and relevant to carry out scientific studies of high scientific merit. Nevertheless, we consider important to highlight this limitation in the Conclusions section. It is also important to highlight that the European projects are accepted by a group of researchers specialized in each of the scientific areas and include researchers from other geographical areas such as North America, Asia, and Oceania. The evaluation is carried out by a panel and the identification of the promoters of each project is omitted. Consequently, all decisions taken to accept projects are based on technical and scientific merit, not on their political convictions.

- Have you analyzed the internal and external validity?

Author’s response: Thanks for your suggestion. We addressed this issue in the methodology section. We have followed external and internal validity procedures. We have considered and presented external validity mechanisms like the representative selection of participants, the comparison of findings to other studies that is performed in the Discussion section, and the transferability that allow to transfer the findings to other situations. As internal validity we have presented the data collection methods applied in this study, the temporal sequence of the process, the use of triangulation considering the abstract and objectives of each project, and we have also identified and presented potential source of bias that are addressed in the Conclusions section.

- In CORDIS, the full project proposal is not available. How did you solve this situation? How are you sure the short description are you considering is a good project descrition? Did you employed any text quality indicator to reject those sentence or paragraph with a poor quality? 

Author’s response: Thanks for your comment. It is true that the full project proposal is not available on the Cordis portal. However, the full project proposal would not be a good source of information due to its large size and lack of a uniform format. On the contrary, the information on the objectives and technical description of the projects is common to all projects and its completion is the responsibility of the promoters. It is expected, not least because the project proposals have been reviewed by an independent panel, that this information is sufficiently rich and of technical and scientific quality. On average this information amounts to around 500 to 750 words for each project. This means that in total this information represents between 31,000 words and nearly 47,000 words. Manual topic identification on this amount of work is very time-consuming and error-prone. For this reason, the analysis was carried out using NVivo software. The various stages of using thematic analysis are shown in Table 2.

Although your methodology is qualitative, for sure you employed techniques to identify synonyms, expressions with an equivalent meaning, etc. This is not described in your paper.

Author’s response: Thanks for your pertinent recommendation. We have included this information in Table 2 that presents the several phases of the thematic analysis process.

Results and conclusions cannot be considered strong until all this information is provided and can the analyzed. Please, update your paper accordingly.

Author’s response: The greater detail in the methodological process led us to have richer information about the results. The former Figure 1 is replaced by Table 3, which presents the most relevant terms identified in the field. We also consider that the reformulation in the presentation of the results also contributed to improve the discussion of our findings. Finally, the conclusions section is also more focused to highlight the main principles of the study.

Reviewer 3 Report

The article deals with the currently very important issue of the functioning of smart cities, and more specifically, protection against cyber-threats that occur in them. The analysis of the topic was based on a wide range of publications including 77 items. These publications show the scope of research currently being carried out in 62 European countries. These works are the result of research, the end of which is expected in 2027. It is to be hoped that, as suggested in the summary, there will be further research on this issue.

Notes on the editing part of the article:

Ø  Figure 1 is redundant as it practically does not introduce anything significant from the point of view of the substantive content of the article.

Ø  It would be recommended to introduce a spacing between rows in the tables, which would increase their readability.

Author Response

Response to Reviewer

We appreciate the review suggestions and comments received by the reviewer. These elements are key to improving the final quality of the manuscript. Below we respond to each issue raised.

Review #3

The article deals with the currently very important issue of the functioning of smart cities, and more specifically, protection against cyber-threats that occur in them. The analysis of the topic was based on a wide range of publications including 77 items. These publications show the scope of research currently being carried out in 62 European countries. These works are the result of research, the end of which is expected in 2027. It is to be hoped that, as suggested in the summary, there will be further research on this issue.

Author’s response: We appreciate the excellent work of synthesizing the objectives and relevance of this study and the contributions given to improve the manuscript.

Notes on the editing part of the article:

- Figure 1 is redundant as it practically does not introduce anything significant from the point of view of the substantive content of the article.

Author’s response: We agree with this reviewer and, therefore, we have deleted this figure. In turn, we present in Table 3 the top 10 frequent themes.

- It would be recommended to introduce a spacing between rows in the tables, which would increase their readability.

Author’s response: We reformat the tables accordingly to improve their readability.

Round 2

Reviewer 1 Report

There is an improvement. The author introduced many important notes, and made significant corrections. This ver.2 file reads easier, the incomplete information is not so frequent.

On the other hand, the author agreed with all the remarks but only the reference-oriented goals had been achieved. In the other cases the additions are insufficient, they resolved only a part of discussed between us problems.

The following main problems have to be resolved:

1. It is written in a 'librarian style'. A librarian who correctly finds the proper keywords from the scope but cannot represent necessary interconnections, and make a deep analysis based on the considered data/keywords/etc. Please introduce more IT experience especially in the data science/machine learning/intelligent fields.

2. Maybe the problem with a rather shallow analysis is because only ad-type texts were analyzed by the author. For example, only CORDIS-based abstracts. If so, the significance of this article is low.

I cannot recommend it for publication in this form.

In my opinion, the methodology/experimental parts should be significantly improved.

Author Response

We appreciate the review suggestions and comments received by the reviewer. These elements are key to improving the final quality of the manuscript. Below we respond to each issue raised.

Review #1

There is an improvement. The author introduced many important notes, and made significant corrections. This ver.2 file reads easier, the incomplete information is not so frequent.

On the other hand, the author agreed with all the remarks but only the reference-oriented goals had been achieved. In the other cases the additions are insufficient, they resolved only a part of discussed between us problems.

Author’s response: Thank you for your general appreciation of the manuscript and for indicating improvements in the work. We remain willing to improve the study and follow your recommendations for improvement.

The following main problems have to be resolved:

  1. It is written in a 'librarian style'. A librarian who correctly finds the proper keywords from the scope but cannot represent necessary interconnections, and make a deep analysis based on the considered data/keywords/etc. Please introduce more IT experience especially in the data science/machine learning/intelligent fields.

Author’s response: Thanks for your recommendation. We have made significant improvements in the adoption of technical terminology associated with the security risks that emerge in the use of artificial intelligence and machine learning in the context of smart cities. The literature review in the field was revised and expanded, and the discussion of the results was also improved. We also consider that there has been a significant evolution of the scientific component of the topic. To this end, the following new references have been included:

Sarker, I.H. Data Science and Analytics: An Overview from Data-Driven Smart Computing, Decision-Making and Applica-tions Perspective. SN Comp. Science 2021, 2, 1-22. https://doi.org/10.1007/s42979-021-00765-8

Xu, Z.; Tang, N.; Xu, C.; Cheng, X. Data science: connotation, methods, technologies, and development. Data Science Manag. 2021, 1, 32-37. https://doi.org/10.1016/j.dsm.2021.02.002

Hazim, K.; Khan, A.M. Cloud Computing: Revolution of the Internet. Int. J. Eng. Res. Tech. 2014, 2, 236-239. https://doi.org/10.17577/IJERTCONV2IS03033

Brataas, G.; Herbst, N.; Ivansek, S.; Polutnik, J. Scalability Analysis of Cloud Software Services. In Proc. of the IEEE Interna-tional Conference on Autonomic Computing (ICAC), Columbus, OH, USA, 2017, 285-292. https://doi.org/10.1109/ICAC.2017.34

Biswas, A.; Majumdar, S.; Nandy, B.; El-Haraki, A. A hybrid auto-scaling technique for clouds processing applications with service level agreements. J. Cloud Comp. 2017, 6, 1-22. https://doi.org/10.1186/s13677-017-0100-5

Soni, D.; Kumar, N. Machine learning techniques in emerging cloud computing integrated paradigms: A survey and taxonomy. J. Netw. Comp. Applic. 2022, 205, 1-39. https://doi.org/10.1016/j.jnca.2022.103419

Peng, Z.; Gong, Q.; Duan, Y.; Wang, Y. The Research of the Parallel Computing Development from the Angle of Cloud Computing. J. Phys.: Conf. Ser. 2017, 910, 1-8. https://doi.org/10.1088/1742-6596/910/1/012002

Syrmakesis, A.D.; Alcaraz, C.; Hatziargyriou, N.D. Classifying resilience approaches for protecting smart grids against cyber threats. Int. J. Inform. Sec. 2022, 21, 1189-1210. https://doi.org/10.1007/s10207-022-00594-7

Huang, Z.; Chen, J.; Lin, Y.; You, P.; Peng, Y. Minimizing data redundancy for high reliable cloud storage systems. Comp. Netw. 2015, 81, 164-177. https://doi.org/10.1016/j.comnet.2015.02.013

Kosmowski, K.T.; Piesik, E.; Piesik, J.; Sliwinski, M. Integrated Functional Safety and Cybersecurity Evaluation in a Framework for Business Continuity Management. Energies 2022, 15, 1-21. https://doi.org/10.3390/en15103610

Burlacu, M.; Boboc, R.G.; Butila, E.V. Smart Cities and Transportation: Reviewing the Scientific Character of the Theories. Sustainability 2022, 14, 1-15. https://doi.org/10.3390/su14138109

Almihat, M.; Kahn, M.; Aboalez, K.; Almaktoof, A. Energy and Sustainable Development in Smart Cities: An Overview. Smart Cities 2022, 5, 1389-1408. https://doi.org/10.3390/smartcities5040071

Cremer, F.; Sheehan, B.; Fortmann, M.; Kia, A.N.; Mullins, M.; Murphy, F.; Materne, S. Cyber risk and cybersecurity: a sys-tematic review of data availability. Geneva Pap. Risk Insur. Issues Pract. 2022, 47, 698-736. https://doi.org/10.1057%2Fs41288-022-00266-6

Ismagilova, E.; Hughes, L.; Rana, N.P.; Dwivedi, Y. Security, Privacy and Risks Within Smart Cities: Literature Review and Development of a Smart City Interaction Framework. Inf. Syst, Front. 2022, 24, 393-414. https://doi.org/10.1007%2Fs10796-020-10044-1

Anthi, E.; Williams, L.; Rhode, M.; Burnap, P.; Wedgbury, A. Adversarial attacks on machine learning cybersecurity defences in Industrial Control Systems. J. Inform. Sec. Applic. 2021, 58, 1-9. https://doi.org/10.1016/j.jisa.2020.102717

Anwar, R.W.; Ali, S. Smart Cities Security Threat Landscape: A Review. Comp. Inform. 2022, 41, 405-423. https://doi.org/10.31577/cai_2022_2_405

  1. Maybe the problem with a rather shallow analysis is because only ad-type texts were analyzed by the author. For example, only CORDIS-based abstracts. If so, the significance of this article is low. I cannot recommend it for publication in this form.

Author’s response: Thanks for the opportunity to clarify. We have described the data preparation phase in Table 2. The data collected on the CORDIS platform for each project is compiled into an individual report. Information on project name, implementation dates, summary, objectives, and technical description are extracted. These elements are available in “fact sheet” and “reporting” sections on CORDIS platform. A script written in Python was developed to automatically collect this information. There aren’t ad-type texts because all the text is written by the promotors of each project. The goal of the CORDIS portal is to provide technical and scientific information on the projects and not to implement a promotional policy because all projects registered on the platform are already funded and are not looking for another source of complementary funding. It is also important to note that the projects identified in this study are all in the development phase, hence there is no project report. Nevertheless, we consider that the information is rich enough to gain valuable insights considering the objectives and technical description of each project that is comprehensive enough regarding the main approaches and technologies used to mitigate cybersecurity risks in smart cities.

In my opinion, the methodology/experimental parts should be significantly improved.

Author’s response: Thanks for your recommendation. We have improved the methodology/experimental part by detailing the Data preparation phase, which is a key phase for the thematic analysis. Initially, a search was made considering all projects funded by Horizon Europe in the smart cities field. This resulted in the identification of 180 projects. After that, the combination of the three words "smart" and "city" and "security" was applied. It was also assured that the current state of development of the projects is relatively similar. Therefore, all projects in the approval phase, or approved but not yet started, have been excluded. In total, and after applying the listed restrictions, a total of 62 projects were identified as presented in Table 1. Associated with each project is the date of its implementation (i.e., start date and end date) and a brief description of its main goal. The identification of the goal of each project was manually carried out considering the main objectives of each project, since the information available in the CORDIS portal is more extensive. Finally, the various phases of the thematic analysis are presented in Table 2. It includes the following phases: data preparation, codification, identification of initial themes, review of themes, definition and naming of themes, analysis of themes, interpretation of results, and reporting of results.

Reviewer 2 Report

In my opinion, the authors have addressed all my previous concerns and the paper may be accepted

Author Response

We appreciate the review suggestions and comments received by the reviewer. These elements are key to improving the final quality of the manuscript. Below we respond to each issue raised.

Review #2

In my opinion, the authors have addressed all my previous concerns and the paper may be accepted.

Author’s response: Thank you very much for your feedback and recommendation to accept this paper.

Round 3

Reviewer 1 Report

The presented article considers a result analysis concerning European smart city cybersecurity issues.

The overview part includes 100+ quotations, it gives an original view to current problems, research, and policies in this rapidly developing field.

The readers could be interested in further analysis of this perspective research, for example, what architectures of Smart city cybersecurity ecosystems are widely applied in the quoted projects.

Some parts of this extended research seem to be rather isolated from the main research. It would be great if the research team will make a more smooth transition between main article topics 

Author Response

We thank the reviewer for his further revision suggestions and for his helpful and collaborative work in this process. Below we respond to each issue raised.

Review #1

The presented article considers a result analysis concerning European smart city cybersecurity issues. The overview part includes 100+ quotations, it gives an original view to current problems, research, and policies in this rapidly developing field.

Author’s response: Thank you very much for your analysis of this paper.

The readers could be interested in further analysis of this perspective research, for example, what architectures of Smart city cybersecurity ecosystems are widely applied in the quoted projects.

Author’s response: Thanks for your suggestion. It is a pertinent observation. We have updated the Results section to present this observation. We identified some common architectural components for the development of smart cities such as the exploitation of information generated through data analytics techniques, the evolution of artificial intelligence that enables a more personalized interaction with citizens, information privacy, and the role that the IoT can assume in various areas such as urban mobility, energy efficiency, and sustainability.

Some parts of this extended research seem to be rather isolated from the main research. It would be great if the research team will make a more smooth transition between main article topics.

Author’s response: Thanks for your recommendation. We have restructured the Introduction section to explicitly present the Research Questions. This approach will help to better guide the author in the discussion of the results. We have also made some improvements in the introduction of some paragraphs to increase the flow in the presentation of the information.